# Dispositional cognitive effort investment and behavioral demand avoidance: Are they related?

**Alexander Strobel**[1][ʘ]*, **Gesine Wieder**[1], **Philipp C. Paulus**[1,2], **Florian Ott**[1], **Sebastian Pannasch**[1], **Stefan J. Kiebel**[1], **Corinna Kührt**[1][ʘ]

**1** Faculty of Psychology, Technische Universität Dresden, Dresden, Germany, **2** Max Planck Institute for Human Cognitive and Brain Sciences, Leipzig, Germany

ʘ These authors contributed equally to this work.
* alexander.strobel@tu-dresden.de

**Data Availability Statement:** All data and materials for reproducing our analyses are permanently and openly accessible at https://osf.io/9thqb.

## Abstract

Individuals tend to avoid cognitive demand, yet, individual differences appear to exist. Recent evidence from two studies suggests that individuals high in the personality traits Self-Control and Need for Cognition that are related to the broader construct Cognitive Effort Investment are less prone to avoid cognitive demand and show less effort discounting. These findings suggest that cost-benefit models of decision-making that integrate the costs due to effort should consider individual differences in the willingness to exert mental effort. However, to date, there are almost no replication attempts of the above findings. For the present conceptual replication, we concentrated on the avoidance of cognitive demand and used a longitudinal design and latent state-trait modeling. This approach enabled us to separate the trait-specific variance in our measures of Cognitive Effort Investment and Demand Avoidance that is due to stable, individual differences from the variance that is due to the measurement occasion, the methods used, and measurement error. Doing so allowed us to test the assumption that self-reported Cognitive Effort Investment is related to behavioral Demand Avoidance more directly by relating their trait-like features to each other. In a sample of $N = 217$ participants, we observed both self-reported Cognitive Effort Investment and behavioral Demand Avoidance to exhibit considerable portions of trait variance. However, these trait variances were not significantly related to each other. Thus, our results call into question previous findings of a relationship between self-reported effort investment and demand avoidance. We suggest that novel paradigms are needed to emulate real-world effortful situations and enable better mapping between self-reported measures and behavioral markers of the willingness to exert cognitive effort.

## Introduction

The role of cognitive effort investment in goal-directed behavior has been discussed for long [1, 2], and there has been a renewed interest in this issue in the last decade. Several contemporary theories [e.g., 3, 4] highlight its importance in value-based decision-making. Crucially,

**Funding:** This work was funded by a grant of the German Research Foundation (Deutsche Forschungsgemeinschaft, DFG, https://www.dfg.de; grant number SFB 940/2) to AS, SP and SJK. The funder had no role in study design, data collection and analysis, decision to publish, or preparation of the manuscript.

**Competing interests:** The authors have declared that no competing interests exist.

individual differences in dispositional cognitive effort investment have been identified that systematically relate to actual effort investment in behavioral paradigms designed to challenge an individual's willingness to engage in cognitive effort investment.

Specifically, Westbrook et al. [5] examined the phenomenon of cognitive effort discounting, defined as the subjective cost to perform a cognitively more strenuous level of an n-back relative to a less demanding one in their so-called Cognitive Effort Discounting (COG-ED) paradigm. They found that individuals with higher Need for Cognition (NFC) exhibited lower cognitive effort discounting. Moreover, they observed that lower effort discounting was related to lower delay discounting that is often viewed as an indicator of self-control [6, 7]. In a similar vein, Kool et al. [8] examined the avoidance of cognitive demand imposed by a Demand Selection Task [DST; 9]. In this task, participants have to choose between one of two visual patterns, and upon choice, one of two simple tasks is revealed. Crucially, one pattern is associated with more frequent switching between the two tasks. Kool et al. [8] observed that overall, participants tended to avoid the cognitive demand imposed by task-switching. However, this effect was found to be less pronounced in individuals with higher levels in the personality trait Self-Control and with a lower tendency towards delay discounting. In a replication attempt, Juvina et al. [10] could not corroborate this finding. However, when using an alternative parametrization that takes into account whether participants actually detect that the choice options in the DST are associated with different demand, they found the expected negative correlations between Demand Avoidance and both Self-Control and NFC.

These findings suggest that stable individual differences in personality traits such as NFC and Self-Control can predict the extent to which individuals discount effort and avoid cognitive demand. Furthermore, they raise the question whether behavioral measures of demand avoidance can to some extent also be considered as a trait, i.e., a tendency to act in certain situations in a certain way that is stable across similar situations and across time [11]. If so, then such behavioral measures would qualify as time-stable predictors for real-life outcomes such as self-control failures in everyday life as measured via ecological momentary assessment (e.g., [12]) or relapse from addiction treatment (e.g., [13]). Thus, the aims of the present research were to determine the trait-like nature of both self-report measures of personality traits related to the willingness to invest mental effort and behavioral measures of effort avoidance in order to systematically relate stable individual differences in these measures to each other.

However, if we administer a measure of a construct at one time point, the variance of the resulting measure is composed of variance due to stable individual differences, i.e., trait variance, but also of other sources of variance: variance due to time-fluctuating occasion-specific influences, variance due to the specific form of measurement we chose, and error variance [14]. Therefore, in order to determine the extent to which the relationship between self-report and behavioral measures of effort investment is due to time-stable individual differences, one needs to separate the trait-variance in these measures from other sources of variance such as variance due to time-fluctuating occasion-specific influences, variance due to the specific form of measurement, and error variance. For this purpose, we used latent state-trait modeling [14]. It allows a variance decomposition via the measurement of two constructs–here a personality trait related to the willingness to invest effort and a putative behavioral trait of effort avoidance–via at least two different methods and measurement occasions to establish to what extent these constructs exhibit trait-like features and to determine their trait covariance. This requires to administer two parallel measures—or indicators—of one construct, and the question arises, whether NFC and Self-Control on the one hand and the COG-ED paradigm and the DST on the other overlap to a degree that we can consider them as parallel indicators of overarching traits.

As for the latter question, we originally intended to use the COG-ED paradigm and the DST as indicators. However, we eventually refrained from using the COG-ED paradigm for a number of reasons (for details, see S1 Appendix: Supplementary Methods). The main reason was the rather low correspondence to be expected of the COG-ED and the DST due to their rather different task structures whereas latent variable modeling requires substantially correlated indicators. We therefore employed two versions of the DST as indicators of one overarching construct of Demand Avoidance. As for the former question, we need to take a closer look on the conceptualization of NFC and Self-Control and on the literature on their relationship.

NFC is conceptualized as an individual's "tendency to engage in and enjoy effortful cognitive activity" [15, p. 197], whereas Self-Control refers to the "ability to override or change one's inner responses, as well as to interrupt undesired behavioral tendencies (such as impulses) and refrain from acting on them." [16, p. 274] or "the deliberate, conscious, effortful subset of self-regulation" [17, p. 351], see also Eisenberg et al. [18]. While the trait definitions of NFC and Self-Control seem somewhat different at first glance, they share the aspect of effortful goal pursuit that points to a common core of both constructs.

Ample evidence implicates NFC in the willingness to invest mental effort during goal pursuit across a variety of domains of information processing such as active search, elaboration, evaluation, and recall of information as well as decision-making and problem-solving [for review, see 15]. Electroencephalographic evidence suggests a higher allocation of attentional resources in individuals high in NFC. Enge et al. [19] found individuals high in NFC to exhibit higher amplitudes in event-related potentials that are indicative of bottom-up and top-down attention allocation in a novelty oddball task. Similarly, Mussel et al. [20] observed that in an n-back task, individuals with higher NFC responded to increased cognitive demands with the recruitment of more cognitive resources, indexed via frontal midline theta power, than individuals with lower NFC levels NFC. Moreover, relating NFC to other personality variables, Fleischhauer et al. [21] found NFC to be associated with traits characterized by openness to experience, activity, achievement-striving, persistence, and drive. Taken together, these strands of evidence underscore the role of NFC in effort investment across multiple domains.

Self-Control can be understood as the capacity to exert effortful control over dominant behavioral tendencies in the pursuit of long-term goals [22]. As example for its role in goal pursuit, Tangney et al. [16] showed that undergraduate students with high scores in self-reported Self-Control produced on average better grades compared to those low in Self-Control and also reported a lower incidence of dysfunctional and impulsive behaviors. Another example comes from a meta-analysis by Hagger et al. [22] on the ego depletion effect, i.e., declining task performance over time due to impaired Self-Control resources, revealed medium to large overall effects on effort, self-reported difficulty and lack of energy. This—albeit not undisputed—evidence suggests that exerting Self-Control is perceived as effortful. As a third example, Lindner et al. [23] demonstrated a positive association of trait Self-Control on the relation of state Self-Control and self-rated effort investment during a 140-minute achievement test in mathematics and science. High trait self-control capacity supported participants to keep state self-control at a higher level, resulting in more effort invest in test-taking. Taken together, Self-Control is a relevant trait that modulates effort investment to achieve goals.

Given the conceptual overlap concerning effort investment, there have been empirical efforts to relate NFC to Self-Control that typically yielded correlations of $r$ = .30 –.40 [24–27]. In one of these studies that preceded the present one, we also aimed to relate both constructs more systematically to each other and to establish a hierarchical factor model. The shared variance of NFC and the conceptually related Intellect scale by Mussel [28] gave rise to a first-

order factor *Cognitive Motivation*. Likewise, the shared variance of Self-Control and the related scale Effortful Control from the Adult Temperament Questionnaire [29] was captured by a first-order factor *Effortful Self-Control*. Crucially, the shared variance of these two first-order factors was explained by a second-order factor *Cognitive Effort Investment*. Thus, NFC and Self-Control can be integrated into a hierarchical model that captures the essence of both traits, i.e., effortful processing and goal-orientation, in the superordinate construct of Cognitive Effort Investment.

Taken together, in the present research, we sought to conceptually replicate and extend the findings of Kool et al. [8] and Westbrook et al. [5] that personality traits capturing the dispositional willingness to invest mental effort during goal-pursuit are related to lower effort avoidant behavior. To this end, we employed a latent variable approach and used two personality measures (Cognitive Motivation and Effortful Self-Control) as indicators of a latent variable Cognitive Effort Investment and two DST versions as indicators of a latent variable of Demand Avoidance at two measurement occasions. This enabled us to determine the trait-like nature of both self-reported Cognitive Effort Investment and behavioral Demand Avoidance in order to systematically relate stable individual differences in these measures to each other. This was done to provide further evidence on the role of personality traits related to the willingness to invest mental effort to behavioral measures of effort avoidant behavior in order to provide a basis for using trait-like behavioral measures as predictors for real-life outcomes. We also controlled for cognitive functioning via a cognitive task battery measuring basic cognitive functions, assuming that individual differences in cognitive ability would be related to the willingness to invest mental effort and therefore to both Cognitive Effort Investment and Demand Avoidance. We hypothesized that even after controlling for cognitive functioning, there would be a negative correlation between the trait aspects of Cognitive Effort Investment and Demand Avoidance.

## Methods

We report how we determined our sample size, all data exclusions, all manipulations, and all measures in the study. All data and materials for reproducing our analyses are permanently and openly accessible at https://osf.io/9thqb. The study was not preregistered.

### Participants

Our sample size calculation was mainly based on a trade-off between pragmatic reasons and statistical affordances. We therefore aimed at a sample size of $N \geq 200$ participants, i.e., a manageable sample size adequate for structural equation modeling and sufficiently powered to detect correlations of $r \geq .20$ at $\alpha = .05$ (two-sided) and $1\text{-}\beta = .80$. To achieve a minimum $N$ of 200, we oversampled by 10% assuming potential dropouts, exclusions and outliers.

Participants were recruited from the local university and were screened via phone to allow for inclusion. Inclusion criteria were age 18–38 years, fluent German language skills, normal or corrected-to-normal vision. Exclusion criteria were any pre-existing psychological, psychiatric, or neurological conditions, regular intake of illegal drugs or excessive intake of legal drugs, and regular intake of medication that could impair mental capacities. Out of a total of 282 volunteers originally screened, 65 could not be included in the final analyses based on either the above criteria, conflicting schedules that did not allow for assessment, failed data recordings or non-compliance to the instructions during assessment (see S1 Appendix: Supplementary Methods). Thus, the final sample comprised $N = 217$ participants (72.4% women, age range 18–39 years, $M = 23.2$, $SD = 4.3$ years). Educational level of the participants was high with 99.5% holding a university entrance diploma and 86.6% being students. The majority of

the sample (91.7%) was right-handed as determined using the Edinburgh Handedness Inventory [30]. Please note that this sample is a subsample of the sample used in Study 2 in Kührt et al. [26] where we replicated the hierarchical factor model of Cognitive Effort Investment. The present sample consists of those participants who took part in both assessments and had complete data for all measures relevant for the present report (for details, see S1 Appendix: Supplementary Methods). There is no duplicate reporting of results.

## Material

**Self-report measures.** In order to have two measures each for personality traits pertaining to Cognitive Motivation and Effortful Self-Control, we employed the following four questionnaire measures:

*Need for Cognition* was assessed with the 16-item short version of the German NFC scale [31]. Responses to each item (e.g., "Thinking is not my idea of fun", recoded) were recorded on a 7-point Likert scale ranging from -3 (completely disagree) to +3 (completely agree). The scale shows comparably high internal consistency of Cronbach's $\alpha > .80$ [21, 31] and a retest reliability of $r_{tt} = .83$ across 8 to 18 weeks [32].

To measure *Intellect*, we employed the Intellect scale by Mussel [28]. It has 24 items to assess individual differences in the two intellectual processes *Seek* and *Conquer* and the three intellectual operations *Think*, *Learn*, and *Create*. The combination of each process and operation gives six facets of Intellect that are measured by 4 items each (e.g., "I enjoy solving complex problems" for the Seek/Think facet or "When I'm developing something new, I can't rest until it's completed" for the Conquer/Create facet). Items are rated on a 7-point Likert scale ranging from -3 (strongly disagree) to +3 (strongly agree). Internal consistency is high, with Cronbach's $\alpha = .94$ for the total Intellect score and $\geq .86$ for the six facets [28].

*Self-Control* was measured using the short form of the German Self-Control Scale [SCS-K-D; 33] that comprises 13 items (e.g., "I am able to work effectively toward long-term goals") with a 5-point Likert scale ranging from -2 (completely disagree) to +2 (completely agree). The scale shows high reliability, Cronbach's $\alpha \sim .80$, 7-week retest reliability of $r_{tt} = .82$ [33].

*Effortful Control* was assessed with the respective scale of the German Adult Temperament Questionnaire [ATQ; 34] that comprises 19 items on executive control in everyday life. Responses to items (e.g., "Even when I feel energized, I can usually sit still without much trouble if it's necessary") are given on a 7-point rating scale from -3 (completely disagree) to +3 (completely agree). With Cronbach's $\alpha = .74$, internal consistency of the scale is acceptable [34].

Scale internal consistencies and retest reliabilities in the present study are given in Table 1 together with descriptive statistics of each self-report measure. We also used two further questionnaires: the German short version of the Big Five Inventory [35] and the German Generalized Self-Efficacy scale [36]. Given the scope of the present research, these questionnaires were not further examined here.

In addition to the personality questionnaires, we employed a measure of perceived task load, the NASA Task Load Index [NASA-TLX; 37] that was administered after the behavioral tasks (see Procedure). In the NASA-TLX, participants evaluate their subjective perception of the mental, physical and temporal demands of a particular task, as well as their performance, effort and frustration during the task on a 20-point scale for each dimension. In its original, the NASA-TLX also requires comparisons of two dimensions each. For this study, we relinquished the comparison due to time restrictions.

**Behavioral tasks.** Two versions of the Demand Selection Task (DST) as introduced by Kool et al. [9] were used. In this type of task, participants are required to choose between two

**Table 1. Spearman correlations and descriptive statistics of the personality measures.**

| | 1 | 2 | 3 | 4 | 5 | 6 | 7 | 8 |
|---|---|---|---|---|---|---|---|---|
| 1 Need for Cognition T1 | .86 | .63 | .28 | .29 | ***.83*** | .61 | .21 | .32 |
| 2 Intellect T1 | | .92 | .29 | .30 | .62 | ***.77*** | .23 | .30 |
| 3 Self-Control T1 | | | .79 | .61 | .31 | .35 | ***.81*** | .62 |
| 4 Effortful Control T1 | | | | .77 | .32 | .34 | .58 | ***.80*** |
| 5 Need for Cognition T2 | | | | | .88 | .70 | .27 | .39 |
| 6 Intellect T2 | | | | | | .93 | .37 | .42 |
| 7 Self-Control T2 | | | | | | | .82 | .72 |
| 8 Effortful Control T2 | | | | | | | | .78 |
| Mean | 16.42 | 24.04 | 1.85 | 8.92 | 14.80 | 22.33 | 1.86 | 8.97 |
| SD | 11.71 | 16.58 | 7.18 | 12.99 | 12.07 | 18.09 | 7.43 | 12.69 |
| Min | -28 | -15 | -14 | -25 | -24 | -32 | -16 | -24 |
| Max | 47 | 63 | 19 | 45 | 47 | 69 | 23 | 39 |
| Skew | -0.48 | -0.28 | 0.17 | 0.04 | -0.62 | -0.23 | 0.11 | 0.08 |
| Kurtosis | 0.86 | -0.22 | -0.60 | -0.06 | 0.61 | 0.20 | -0.59 | -0.31 |

$N$ = 217; all coefficients significant at $p \leq .002$; coefficients in the diagonal are Cronbach's $\alpha$, bold-faced coefficients give the 5-week retest reliability; T1 and T2 denote the measurement occasions 1 and 2; approximated standard errors for skew and kurtosis are 0.17 and 0.33.

visual patterns. Upon choosing one pattern, the actual task is revealed. It requires participants to evaluate whether a given single-digit numeral is less or greater than 5, if the numeral has a certain color, or whether the numeral is odd or even, if the numeral has a certain other color. Crucially, one of the two patterns is associated with the same color of the numerals in 90% of choices, i.e., requires to indicate whether the numeral is less or greater than five. The other pattern is associated with the same color in only 10% of choices, resulting in frequent task switching if that pattern is chosen because alternating judgments are required. The basic idea here is that participants would tend to avoid the cognitive demand imposed by frequent task switching and therefore would choose the pattern associated with less demand, i.e., the one associated with less task switching more often. We closely followed the original instruction of Kool et al. [9] that informed the participants as follows: "Subjects were told that they were free to choose from either deck on any trial and that they should 'feel free to move from one deck to the other whenever you choose' but also that 'if one deck begins to seem preferable, feel free to choose that deck more often.'" [9]. Pilot studies had shown the relevance of the instruction's second part because a more neutral instruction did not result in a higher frequency of low demand choices. This is consistent with evidence provided by Juvina and colleagues [10] where in Experiment 1, demand avoidance was less pronounced when using a more neutral instruction that did not bias for demand selection. Our first DST version required a magnitude/parity (MP) evaluation as in Kool et al. [9, Study 1] and Kool et al. [8]. The second DST version required a sound/orthography (SO) evaluation, i.e. to indicate whether a given letter was a vowel or consonant vs. an uppercase or lowercase letter. Task order was randomly assigned to the participants. For both tasks, there was an initial training period comprising a maximum of 8 blocks of 24 trials each that ensured 80% accuracy in the DST categorization. Both tasks were then delivered in eight blocks comprising 72 trials with a variable trial duration depending on the participants' choice and reaction time (with the latter being limited to 1 second) and no inter-trial-interval. At the end of each block of the main task, participants received feedback on the percentage of correct answers for this block and a notification of either appraisal or a request to try harder. Over trials, reaction times, the error rate and the frequency of easy task

choices were recorded. Internal consistencies of different DST variants can be considered as high with Cronbach's $\alpha$ ranging from .85 to .93 [9].

**Cognitive task battery.** As cognitive abilities such as processing speed or switching ability may have an impact on the choice behavior in the demand selection task, we employed a short cognitive task battery at the beginning of each appointment. The battery comprised five tasks of which we eventually only used the Trail-Making Test A and B [38], see S1 Appendix: Supplementary Methods for details on the other tasks. In the Trail Making Test A, 25 numbers scattered across a sheet of paper are to be connected in ascending order. In version B, the task is to connect numbers and letters in alternating order (i.e., 1-A-2-B etc.). The outcome measure is the time for completing the tasks in seconds. Thus, these tasks allow to examine mental speed and task shifting ability.

## Procedure

The study protocol was approved by the ethics committee of the Technische Universität Dresden (reference number EK3012016). Prior to testing, written informed consent was obtained. Data were collected at two time points with an interval of five weeks, although some minor deviations from this schedule occurred (range of days between measurement occasions 34–49, median = 35 days) due to time constraints of the participants. Assessments were taken at the same weekday and the same time of day. At each appointment, up to three participants were tested in parallel with a time lag of 30 minutes to increase efficiency of testing. Participants received 8 € per hour invested for each appointment. Furthermore, participants had the chance to receive an additional 10 € bonus if they completed both appointments. Both appointments had the same setup and measures except for the first measurement occasion, when participants had their color vision tested and were asked to complete a short sociodemographic questionnaire as well as the EHI to determine handedness. The study protocol comprised four blocks: A cognitive task battery testing for individual working memory capacity, processing speed and shifting ability formed the first block of assessment. This took approx. 10–20 minutes. In the second block, participants worked for approx. 30 minutes on one version of the DST. The third block comprised personality questionnaires that took about 20–30 minutes and the fourth block comprised the other version of the DST. The DST versions were presented in random order. After each block, self-perceived effort in the respective task was assessed with the NASA-TLX.

## Statistical analysis

We used *RStudio* [version 1.1.463; 39] with *R* [version 3.5.2; 40]for statistical analyses, with the main analyses carried out using the packages *psych* [version 1.8.12; 41] and *lavaan* [version 0.6.5; 42], see S1 Appendix: Supplementary Methods for all packages employed. All measures in the study were initially analyzed with regard to descriptive statistics, reliability (retest-reliability $r_{tt}$ as well as Cronbach's $\alpha$ and MacDonald's $\omega$ where applicable), and possible deviation from univariate normality as determined via Shapiro-Wilks tests with a threshold of $\alpha$ = .20. Possible differences between the measurement occasions T1 and T2 were descriptively assessed via boxplots, with overlapping notches—that can roughly be interpreted as 95% confidence intervals of a given median—pointing to noteworthy differences. This assessment suggested that no formal statistical difference tests were necessary.

Correlation analyses were performed using Pearson correlations or Spearman correlations if the majority of the variables ($>$ 50%) deviated from univariate normality. Where appropriate, evaluation of statistical significance was based on uncorrected *p*-values or 95% confidence intervals (CI) and evaluation of effect sizes was based on the empirical guidelines provided by

Gignac and Szodorai [43] who—judging from the distribution of correlation coefficients found among psychological studies included in meta-analytic reviews—suggested to categorize correlations as small for $r < .20$, as medium for $.20 \leq r \leq .30$, and as large for $r > .30$. Given the sample size of $N = 217$, we had a power $1\text{-}\beta = .84$ to detect at least medium-sized correlations at a significance level of $\alpha = .05$.

In a next step, we computed the indicator variables for latent state-trait modeling. For the *personality measures*, we used confirmatory factor analysis (CFA) to derive factor scores for the first-order factors of the hierarchical factor model of Cognitive Effort Investment established in Kührt et al. [26]. In this model, Cognitive Effort Investment forms a second-order factor that explains the shared variance of the first-order factors Cognitive Motivation, being estimated from the indicator variables NFC and Trait Intellect, and Effortful Self-Control, being estimated from the indicator variables Self-Control and Effortful Control. Here, we used this model to estimate the individual scores on the latent variables Cognitive Motivation and Effortful Self-Control at T1 and T2 in order to use these as indicator variables for latent state Cognitive Effort Investment at both time points. CFA model specification included free, but equal loadings of the defining variables of a given factor, equal residuals of the first-order factors and equal error variances of the indicator variables. As the model included both time points, the error variances of each indicator variable at T1 was allowed to correlate with the error variance of the same indicator variable at T2, as were the variances of the higher-order factors. Because Mardia tests indicated that the raw questionnaire scores deviated from multivariate normality, $p_{skew} = .012$, $p_{kurtosis} < .001$, these variables were normalized using Blom's formula $(r-3/8)/(n+1/4)$, with $r$ being the rank of observations and $n$ the sample size [44]. Because there was still some deviation from multivariate normality in the normalized data set, $p_{skew} = .420$, $p_{kurtosis} < .001$, we used robust Maximum Likelihood (MLR) for parameter estimation. Model fit was evaluated via the comparative fit index (CFI), root mean square error of approximation (RMSEA), and standardized root mean square residual (SRMR), with values of CFI $\geq .95$, RMSEA $\leq .06$, and SRMR $\leq 0.08$ indicating good model fit [45].

For the *behavioral data of the DST*, we first extracted the percentage of easy demand choices for each block except the training blocks of both tasks at both time points to calculate internal consistencies. Then we used the overall demand avoidance, i.e., the percentage of easy demand choices across all blocks of each task [8, 9] for the latent state-trait models (see S1 Fig in S1 Appendix: Supplementary Results) for exemplary choice patterns of eight randomly selected participants). Because of a recent report stating that in the DST, demand selection depends on whether participants actually detect that the choice options are associated with different demand [10], we also calculated a new demand avoidance measure that is based on both a demand detection point, i.e., the trial number after which a consistent choice pattern emerges, and the percentage of low demand choices after that point (see S1 Appendix: Supplementary Methods).

With regard to the *cognitive task battery*, the variables used were the times needed for completing the Trail Making Test version A and B, while the other cognitive tasks employed were discarded (see S1 Appendix: Supplementary Methods for details). The respective scores were highly correlated at both measurement occasions, $r \geq .51$. Given that these measures capture individual differences in mental speed, task shifting and to some extent also working memory capacity, we considered these measures as sufficiently general measures of cognitive functioning.

We also examined whether demand avoidance and cognitive functioning would be related to *task load*. To avoid a large number of significance tests, we averaged the NASA-TLX scores on the dimensions mental and time demand as well as invested effort separately for each assessment during the experiment and related them to the respective measures in the preceding tasks, i.e., demand avoidance and cognitive ability.

The main analyses comprised the *latent state-trait modeling*. Our primary goal was to comprehensively test the assumption that individuals who are more willing to invest mental effort would show less demand avoidance. To this end, we used the CFA-derived factor scores as measures of personality and modeled them together with the demand avoidance measures and the cognitive function measures separately for the original and the new demand avoidance measure, i.e., two models were fitted, again using normalized data and MLR estimation. Model specification was as follows: Cognitive Motivation and Effortful Self-Control at the two time points T1 and T2 were the indicator variables of latent state Cognitive Effort Investment at T1 and at T2. The latent states Demand Avoidance at T1 and T2 were estimated from the indicator variables pertaining to the two DST variants at the two time points. The two latent Cognitive Functioning states were estimated from the respective scores in the TMT versions A and B. The latent traits Cognitive Effort Investment, Demand Avoidance and Cognitive Functioning were then estimated from the two respective latent states. Furthermore, for each indicator variable, a latent method factor was estimated from the respective scores at the two time points, e.g., the method factor for the Cognitive Motivation measures from the respective scores at T1 and T2. All loadings were fixed to 1, and all variables in the model had an intercept of 0. We imposed the following constraints on our model: For each of the three variable types in the model, i.e., personality, behavioral, and cognitive variables, we assumed equal error variances of the respective four indicator variables and equal latent state residuals of the respective two states. This specification corresponds to the most restrictive model formulated by Steyer et al. [14]. In addition, we assumed equal variances of the two method factors pertaining to each variable type. Furthermore, every latent method factor was specified as being uncorrelated with every other latent variable in the model. We finally regressed latent trait Cognitive Effort Investment and Demand Avoidance on latent trait Cognitive Functioning to control for cognitive ability.

We then defined the variances of the latent states as sums of the variances of the respective latent trait and latent state residuals and the variances of the indicator variables as sum of the variances of the respective latent states, method factors, and errors. From these variances, the four central parameters of latent state-trait theory can be calculated: *Reliability*, i.e., the reliable variance in a given indicator variable, is the sum of the respective state and method factor variances divided by the total variance of the indicator variable. *Trait consistency*, i.e., the variance portion in a given indicator variable that is attributable to stable individual differences in the latent trait, is the variance of the respective latent trait divided by the total variance of the indicator variable. *Occasion specificity*, i.e., the variance portion in an indicator variable that is due to systematic, but unstable differences between individuals at a given measurement occasion, is the latent state residual divided by the total variance of the indicator variable. Finally, *method specificity*, i.e., the variance portion of the indicator variable that is due to non-equivalence of the indicators, is the variance of the respective method factor divided by the indicator variable's variance. Trait consistency, occasion specificity and method specificity sum up to reliability. Note that due to the equality constraints imposed to the model, the estimates of the four parameters are identical for all indicator variables pertaining to each variable type.

## Results

Table 1 gives descriptive statistics for the personality scales as well as their interrelations because Spearman correlations as the majority of the scales showed a non-normal distribution, Shapiro-Wilks tests, $p > .20$. Reliability estimates are provided as well. All measures showed comparably high internal consistencies, Cronbach's $\alpha \geq .77$ and high 5-week retest reliabilities, $r_s \geq .78$. Fig 1A–1D provides boxplots of the personality scales. No noteworthy differences in

the personality measures were observed between T1 and T2. At the first measurement occasion, NFC and the related Intellect scale were correlated with a large effect size, $r_s$ = .63, 95% CI [.54, .70]. The correlation of the Self-Control scale with the related Effortful Control scale had a large effect size as well, $r_s$ = .61, 95% CI [.51, .68]. As expected, NFC showed medium correlations with Self-Control, $r_s$ = .28, 95% CI [.15, .40], and Effortful Control, $r_s$ = .29, 95% CI [.16, .41]. Similar correlations were obtained for the NFC-related Intellect scale. At the second measurement occasion, comparable or even stronger associations were observed (see Table 1).

Next, we performed CFA to derive factor scores of the latent variables Cognitive Motivation and Effortful Self-Control at both time points. Model fit of the assumed hierarchical factor model (see S2 Fig in S1 Appendix: Supplementary Results) was good, $\chi^2$ = 13.22, $df$ = 11, $p$ = .279, CFI = 1.00, RMSEA = .03 with 90% CI [.00, .07], SRMR = .03. The latent variables showed high internal consistencies, MacDonald's $\omega \geq$ .78, and 5-week retest reliabilities $r_s \geq$ .96. Fig 1E and 1F provides boxplots of the latent factor scores.

Fig 1G and 1H gives the boxplots of the scores in the Trail-Making Test versions A and B, with lower scores indicating better performance. At T2, the scores were substantially lower, pointing to learning effects. Nevertheless, 5-week retest reliabilities were substantial, $r_s \geq$ .63. Task load during the cognitive task battery was at best weakly associated with performance in the Trail-Making Tests, $-.23 \leq r_s \leq -.07$.

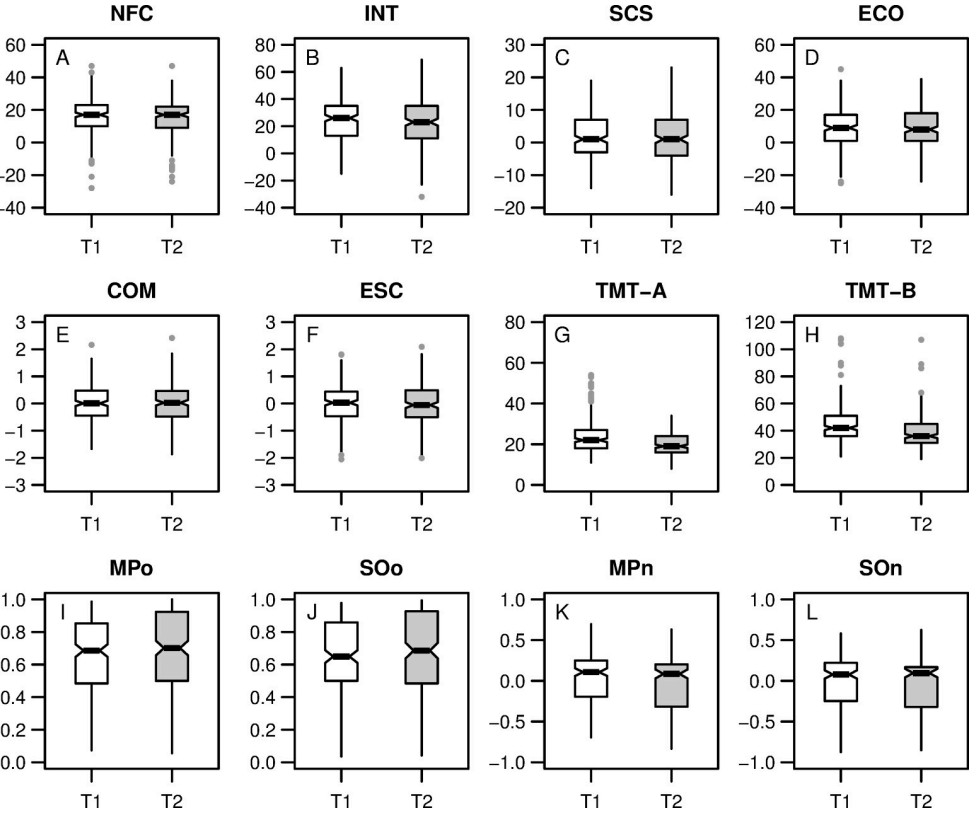

**Fig 1. Boxplots of the variables in the study.** (A-D) raw personality scale scores, NFC = Need for Cognition, INT = Trait Intellect, SCS = Self-Control Scale, ECO = Effortful Control; (E-F) factor scores of COM = Cognitive Motivation and ESC = Effortful Self-Control as derived from confirmatory factor analysis; (G-H) TMT-A/B = Trail-Making Test A and B scores; (I-L) proportion of low demand choices in the two demand selection tasks MP = Magnitude/Parity evaluation and SO = Sound/Orthography evaluation, for o = original and n = new measure.

Table 2 provides the descriptive statistics for the old and new demand avoidance measures together with their interrelations as Spearman correlations due to the non-normal distribution of all behavioral variables, Shapiro-Wilks tests, $p > .20$. The demand avoidance measures showed acceptable to high internal consistencies, especially at T2, Cronbach's $\alpha \geq .70$. Five-week retest reliabilities acceptable, $r_s \geq .57$. We observed the expected pattern of choice behavior, i.e., participants tended to choose the lower demand option more often, both for the original demand avoidance measure (see Fig 1I and 1J) and—less pronounced—for the new demand avoidance measure (see Fig 1K and 1L). Both measures were highly correlated, $r_s \geq$ .50 (see S3 Fig in S1 Appendix: Supplementary Results). Self-reported task load during the two DST variants was not related to demand avoidance, $-.10 \leq r_s \leq .03$. However, across time and tasks, demand avoidance was to some extent related to the scores in the Trail-Making Tests, $-.19 \leq r_s \leq -.01$, $.004 \leq p \leq .927$ for the original measure and $-.09 \leq r_s \leq .17$, $.015 \leq p \leq .473$ for the new measure, justifying the inclusion of the cognitive measures as control variables in the latent state-trait model.

With regard to the question whether participants were aware of the different demand associated with the two patterns (or cared about demand at all), an inspection of the individual demand detection points revealed that despite overall rather early demand detection (with a range of median demand detection points of 8.5 to 12), 16–22% of the participants reached a demand detection point only after half of the block, and 2–4% never reached a demand detection point.

To address the possibility that choice behavior in our implementation of the DST was to some extent driven by an error avoidance strategy, we analyzed the data as follows: For each individual, DST version, and time point, we predicted the average choice behavior (original demand avoidance measure only) in blocks two to eight by the average hit rate during the preceding block by means of a linear mixed model, allowing for random intercepts and slopes per individual. Hit rates during the previous block did not significantly predict choice behavior in the current block (all $p > .182$).

Ahead of latent state-trait modeling, we inspected bivariate correlations between the target personality variables of the present report, i.e., Self-Control and NFC, and behavioral Demand

**Table 2. Spearman correlations and descriptive statistics of demand avoidance measures.**

|  | 1 | 2 | 3 | 4 | 5 | 6 | 7 | 8 |
|---|---|---|---|---|---|---|---|---|
| 1 Magnitude/Parity DST original T1 | *.74* | .54 | .58 | .32 | **.61** | .32 | .57 | .27 |
| 2 Magnitude/Parity DST new T1 |  | *.80* | .34 | .69 | .32 | **.55** | .35 | .56 |
| 3 Sound/Orthography DST original T1 |  |  | *.70* | .50 | .61 | .28 | **.61** | .27 |
| 4 Sound/Orthography DST new T1 |  |  |  | *.75* | .32 | .55 | .37 | **.59** |
| 1 Magnitude/Parity DST original T2 |  |  |  |  | *.81* | .54 | .72 | .37 |
| 2 Magnitude/Parity DST new T2 |  |  |  |  |  | *.84* | .43 | .73 |
| 3 Sound/Orthography DST original T2 |  |  |  |  |  |  | *.81* | .56 |
| 4 Sound/Orthography DST new T2 |  |  |  |  |  |  |  | *.83* |
| Mean | 0.66 | 0.02 | 0.67 | 0.00 | 0.67 | -0.03 | 0.66 | -0.04 |
| SD | 0.23 | 0.34 | 0.22 | 0.31 | 0.26 | 0.34 | 0.26 | 0.34 |
| Min | 0.07 | -0.69 | 0.03 | -0.88 | 0.06 | -0.83 | 0.04 | -0.85 |
| Max | 0.99 | 0.70 | 0.98 | 0.58 | 1.00 | 0.63 | 0.99 | 0.63 |
| Skew | -0.35 | -0.49 | -0.34 | -0.49 | -0.48 | -0.49 | -0.41 | -0.48 |
| Kurtosis | -0.91 | -0.68 | -0.69 | -0.51 | -0.84 | -0.66 | -0.89 | -0.66 |

$N = 217$; all coefficients significant at $p \leq .001$; Cronbach's $\alpha$ given in the diagonal, 5-week retest reliability given bold-faced; DST = Demand Selection Task; T1 and T2 denote the measurement occasions 1 and 2.

Avoidance in order to directly test the hypothesis of a relation between personality traits related to Cognitive Effort Investment and Demand Avoidance. Across tasks and measurement occasions, Self-Control scores were not related to Demand Avoidance, $-.06 \leq r_s \leq .09$, $p \geq .168$, for the original measure of Demand Avoidance and $-.01 \leq r_s \leq .09$, $p \geq .194$, for the new measure.

NFC also showed no relation to Demand Avoidance, $-.05 \leq r_s \leq .06$, $p \geq .390$, for the original measure of demand avoidance and $-.05 \leq r_s \leq .03$, $p \geq .461$, for the new measure. Similar results were observed for Intellect and Effortful Control. Despite the null correlations, we nevertheless proceeded with latent state-trait modeling for the following reasons: First, this approach—yielding compound trait variables unattenuated by measurement error, state and method influences—could still result in a small, but significant relation between trait effort investment and trait Demand Avoidance; second, so far we had not partialled out potential influences of cognitive functioning that could in principle have an impact of both habitual effort investment and behavioral Demand Avoidance; and third, the information on the relative portion of trait, state, and method variance in our measures was itself worth a closer examination.

We fitted two latent state-trait models, one with the original demand avoidance measures as behavioral variables (model 1) and one with the new demand avoidance measures (model 2). Otherwise, the model structure was the same, i.e., the personality variables were the CFA-derived factor scores in Cognitive Motivation and Effortful Self-Control, while the variables pertaining to cognitive functioning were the scores in the Trail-Making Test version A and B. Model 1 (see Table 3 for the correlation matrix) showed a good fit to the data, $\chi^2 = 121.72$, $df = 73$, $p = < .001$, CFI = 0.97, RMSEA = .06 with 90% CI [.04, .07], SRMR = .04. Fig 2 depicts the standardized solution and Table 4 gives the coefficients of the parameters of latent state-trait theory, i.e., reliability, trait consistency, occasion specificity, and method specificity.

For Cognitive Effort Investment, we observed a high reliability of .98, i.e., only 2% of the variance in the Cognitive Motivation and Effortful Self-Control factor scores could not be explained by the latent variables in the model. Lower, but still substantial reliability coefficients were obtained for Cognitive Functioning (.69) and Demand Avoidance (.67). Occasion-specific influences were rather low (.14), while quite high method-specific influences were found for Cognitive Effort Investment (.42), reflecting a considerable degree of non-equivalence of

**Table 3. Pearson correlations of the normalized variables used for latent state-trait modeling: Original demand avoidance measure.**

| | 2 | 3 | 4 | 5 | 6 | 7 | 8 | 9 | 10 | 11 | 12 |
|---|---|---|---|---|---|---|---|---|---|---|---|
| 1 Cognitive Motivation T1 | .52 | .06 | .02 | -.03 | -.04 | **.96** | .48 | .06 | -.05 | -.06 | -.06 |
| 2 Effortful Self-Control T1 | | .03 | -.03 | .04 | -.03 | .59 | **.96** | -.02 | -.01 | -.07 | .02 |
| 3 Trail-Making Test A T1 | | | .57 | -.15 | -.11 | .08 | .05 | **.62** | .51 | -.03 | -.14 |
| 4 Trail-Making Test B T1 | | | | -.17 | -.16 | .04 | -.02 | .57 | **.67** | -.07 | -.14 |
| 5 Magnitude/Parity DST T1 | | | | | .59 | -.04 | .02 | -.09 | -.14 | **.60** | .56 |
| 6 Sound/Orthography DST T1 | | | | | | -.05 | -.03 | -.13 | -.11 | .64 | **.64** |
| 7 Cognitive Motivation T2 | | | | | | | .60 | .08 | .00 | -.07 | -.07 |
| 8 Effortful Self-Control T2 | | | | | | | | -.01 | .02 | -.08 | -.01 |
| 9 Trail-Making Test A T2 | | | | | | | | | .60 | -.01 | -.03 |
| 10 Trail-Making Test B T2 | | | | | | | | | | -.03 | -.02 |
| 11 Magnitude/Parity DST T2 | | | | | | | | | | | .71 |
| 12 Sound/Orthography DST T2 | | | | | | | | | | | — |

$N = 217$; $p < .05$ for $|r| > .14$; 5-week retest reliability given bold-faced; DST = Demand Selection Task; T1 and T2 denote the measurement occasions 1 and 2.

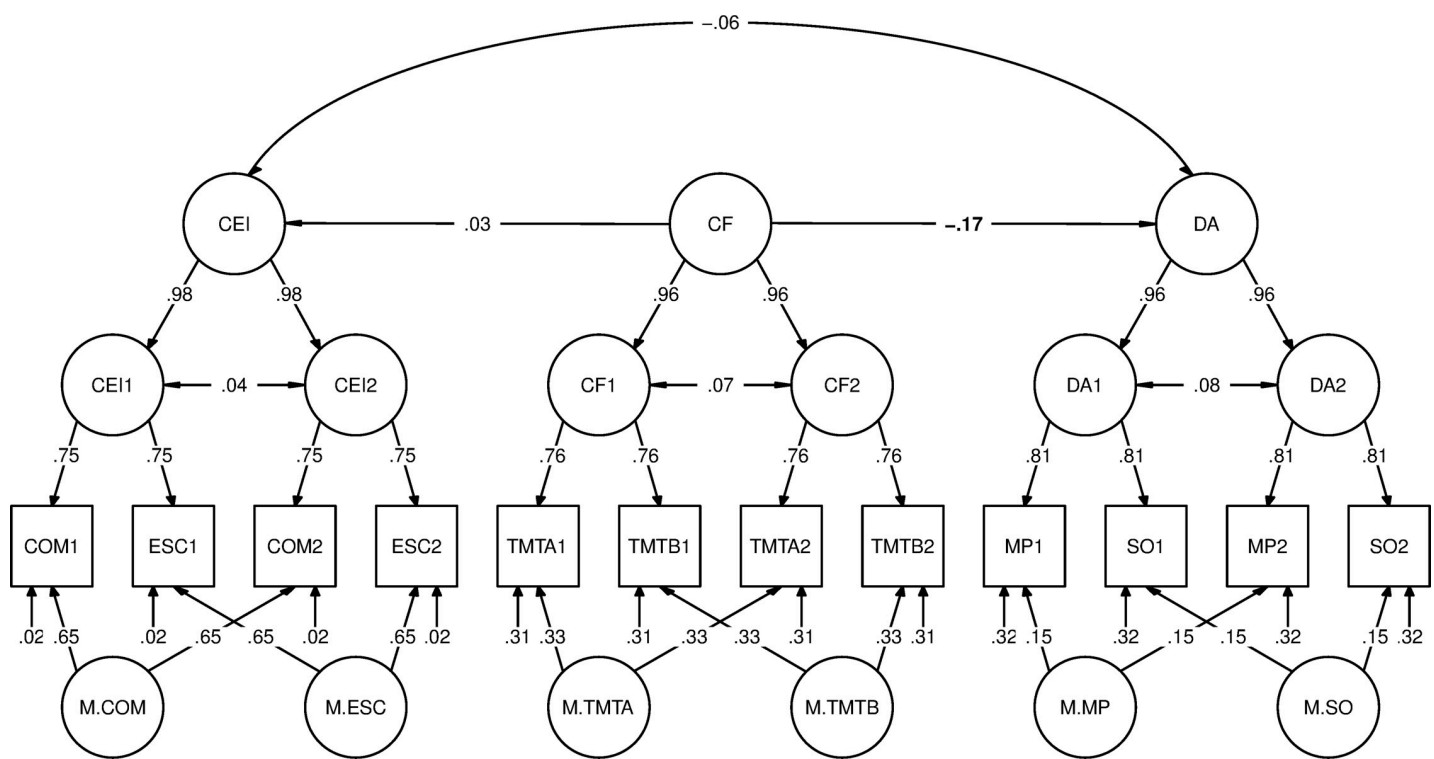

**Fig 2. Latent state-trait model.** Depicted is the relation between trait Cognitive Effort Investment (CEI) and Demand Avoidance (DA), controlled for Cognitive Functioning (CF) at the top as estimated from latent state CEI, DA and CF at the next-lower level (bold-faced: $p < .05$). Indicator variables in squares are COM = Cognitive Motivation and ESC = Effortful Self-Control factor scores, TMT = Trail-Making Test scores in versions A and B, MP = Demand Selection Task with Magnitude/Parity evaluation, SO = Demand Selection Task with Sound/Orthography evaluation, at the two measurement occasions 1 and 2; M = latent method factors at the bottom.

Cognitive Motivation and Effortful Self-Control in measuring Cognitive Effort Investment. Most importantly, the highest portion of variance in all indicator variables was attributable to stable individual differences, with trait consistency estimates ranging from .54 to .59.

However, these stable individual differences were not or only loosely related to each other: While Demand Avoidance was to some extent predicted by Cognitive Functioning, estimate = -0.18, 95% CI [-0.35, -0.01], standardized estimate = -.17, $p = .043$, Cognitive Effort Investment was not, estimate = 0.03, 95% CI [-0.15, 0.22], standardized estimate = .03, $p = .713$, and there was no sizeable covariance between residual Cognitive Effort Investment and Demand Avoidance, estimate = -0.03, 95% CI [-0.14, 0.08], standardized estimate = -.06, $p = .563$.

Model 2 (see Table 5 for the correlation matrix) had a good fit as well, $\chi^2 = 114.73$, $df = 73$, $p = .001$, CFI = 0.98, RMSEA = .05 with 90% CI [.03, .07], SRMR = .04, and yielded similar

Table 4. Parameters of latent state-trait theory.

|  | Cognitive Effort Investment | Cognitive Functioning | Demand Avoidance (original) | Demand Avoidance (new) |
|---|---|---|---|---|
| Reliability | .98 | .69 | .67 | .71 |
| Trait Consistency | .54 | .54 | .59 | .54 |
| Occasion Specificity | .02 | .04 | .05 | .14 |
| Method Specificity | .42 | .11 | .02 | .03 |

$N = 217$.

**Table 5. Pearson correlations of the normalized variables used for latent state-trait modeling: New demand avoidance measure.**

| | 2 | 3 | 4 | 5 | 6 | 7 | 8 | 9 | 10 | 11 | 12 |
|---|---|---|---|---|---|---|---|---|---|---|---|
| 1 Cognitive Motivation T1 | .52 | .06 | .02 | -.07 | -.03 | ***.96*** | .48 | .06 | -.05 | -.07 | -.09 |
| 2 Effortful Self-Control T1 | | .03 | -.03 | .03 | -.04 | .59 | ***.96*** | -.02 | -.01 | -.08 | -.05 |
| 3 Trail-Making Test A T1 | | | .57 | -.08 | -.04 | .08 | .05 | ***.62*** | .51 | .07 | .00 |
| 4 Trail-Making Test B T1 | | | | -.04 | -.04 | .04 | -.02 | .57 | ***.67*** | .10 | -.01 |
| 5 Magnitude/Parity DST T1 | | | | | .67 | -.05 | .03 | .03 | -.08 | ***.53*** | .56 |
| 6 Sound/Orthography DST T1 | | | | | | -.02 | -.04 | .01 | .01 | .53 | ***.61*** |
| 7 Cognitive Motivation T2 | | | | | | | .60 | .08 | .00 | -.05 | -.08 |
| 8 Effortful Self-Control T2 | | | | | | | | -.01 | .02 | -.08 | -.05 |
| 9 Trail-Making Test A T2 | | | | | | | | | .60 | .16 | .06 |
| 10 Trail-Making Test B T2 | | | | | | | | | | .15 | .07 |
| 11 Magnitude/Parity DST T2 | | | | | | | | | | | .70 |
| 12 Sound/Orthography DST T2 | | | | | | | | | | | — |

$N$ = 217; $p$ < .05 for |$r$| > .15; 5-week retest reliability given bold-faced; DST = Demand Selection Task; T1 and T2 denote the measurement occasions 1 and 2.

results. Compared to the original demand avoidance measure, reliability of the new demand avoidance measure was slightly higher (.71), trait consistency was lower (.54), at the expense of a higher occasion specificity (.14). Again, after controlling for Cognitive Functioning, the trait variances of Cognitive Effort Investment and Demand Avoidance were unrelated, estimate = -0.04, 95% CI [-0.15, 0.06], standardized estimate = -.08, p = .395.

Given the near absent correlation between Cognitive Effort Investment and Demand Avoidance, we finally also fitted another model (for the original Demand Avoidance measure only) where we only included the two personality scales related to Effortful Self-Control in order to rule out that the inclusion of measures not directly related to the construct examined by Kool et al. (2013) distorted the personality-behavior relation. While in this case, we obtained a higher trait consistency (.64) and a lower method-specificity (.18) together with comparable occasion-specificity (.05), the trait covariance of Effortful Self-Control and Demand Avoidance was even lower, estimate = -0.01, 95% CI [-0.11, 0.09], standardized estimate = -.01, p = .867.

## Discussion

The present study was conducted in order to conceptually replicate the results by Kool et al. [8] with the two major aims being (1) to assess the extent to which a behavioral measure of the avoidance of cognitive effort is trait-like and (2) to determine whether the trait variance of demand avoidance systematically relates to self-reported cognitive effort investment. To this end, we assessed behavioral demand avoidance and self-reported cognitive effort investment twice within an interval of five weeks and used latent state-trait modeling to separate the trait variance in our measures from occasion- and method specific as well as from error variance to obtain purer measures of demand avoidance and cognitive effort investment. Moreover, both measures were controlled for basic cognitive functioning. While we could show that not only self-reported cognitive effort investment, but also behavioral demand avoidance showed a considerable portion of trait variance, both traits did not covary to a substantial degree. In the following, the results will be integrated into the existing literature, strengths and limitations of the present study will be discussed, and recommendations for future research will be delineated.

## Self-reported cognitive effort investment and behavioral demand avoidance are trait-like

Our results show that more than half of the variance in our measures of self-reported cognitive effort investment (54%) and behavioral demand avoidance (59%) were due to time-stable individual differences. Interestingly, relative to the reliable variance, behavioral demand avoidance even showed a stronger trait component than the self-report measures that also exhibited a higher degree of method variance (39%), attributable to the non-equivalence of the measures for cognitive effort investment. Obviously, in the present study, traits related to Cognitive Motivation were more distinct from traits related to Effortful Self-Control than were the two versions of the demand selection task from each other. Accordingly, when only examining the scales related to Self-Control, trait consistency was higher and method specificity was lower. Still, compared to the literature on latent state-trait analyses [for a comprehensive overview, see 46], the amount of observed trait variance appears substantial. To give a few examples, figural reasoning was found to exhibit about 70% trait variance [47], broad personality traits were reported to show trait variances between 50% and 88% [48], while a narrowly defined trait such as Justice Sensitivity showed a somewhat lower trait variance of about 60% [49]. Thus, our results render our approach as capable of answering the main research question, i.e., to what extent *dispositional* demand avoidance and cognitive effort investment relate to each other.

## Self-reported cognitive effort investment and behavioral demand avoidance are unrelated

The trait variances of self-reported cognitive effort investment and behavioral demand avoidance were not related to each other. This was the case using the standard parametrization of demand avoidance, i.e., the percentage of low demand choices throughout the respective paradigm [8, 9], and a newly proposed parametrization that considers the fact that demand *avoidance* needs to be separated from demand *detection* [10]. Also, when only including personality measures of Self-Control in the model and thus more directly following up on the finding by Kool et al. [8], no relation was obtained. Thus, neither the operationalization of demand avoidance in the DST nor the broader approach to personality traits related to effort investment provides a viable answer for the lack of effects obtained here. How can the absence of the expected effect therefore be explained otherwise? A lack of power to detect such an effect is not an issue here: Kool et al. [8] examined 50 participants and found a correlation between self-reported Self-Control and behavioral demand avoidance of $r = .38$, yielding a power to detect the observed effect at $\alpha = .05$ of $1-\beta = .79$. In comparison, our sample comprised 217 individuals, resulting in an equal power to detect even *half* of the effect size observed by Kool et al. [8]. Another explanation regards the comparability of our sample to that examined by Kool et al. [8]. Yet, both our sample and that of Kool et al. [8] were student samples, and if cultural differences between Germany and the USA would explain the differences, the generalizability of the original finding needed to be questioned. A third possibility could be that we deviated from the original implementation of the DST in some perhaps crucial regard: we gave performance feedback at the end of each block. Therefore, we may not have obtained a pure measure of demand avoidance, because choice behavior could to some extent also have been driven by error avoidance. However, in both versions of the DST at both time points, choice behavior in a given block was not predicted by hit rates in the preceding block. Still, it remains a limitation that we did not establish a task environment identical to that of the original DST. A final explanation may arise from the nature of examined variables, i.e., self-report measures of personality traits and behavioral measures in cognitive tasks, and the approaches taken in personality psychology and cognitive psychology.

## Personality-behavior relationships are weak at best

Evidence for relationships between behavior in executive functioning tasks and personality traits such as those examined here generally points to low or absent direct relationships: In a meta-analysis of the convergent validity of self-control measures [50], the average relation between self-report measures of self-control and executive functioning tasks was $r = .10$. In a study on the relation between NFC with intelligence and working memory, a direct relationship was found for measures of intelligence but not for working memory [51]. Similarly, in a study from our lab, we could not establish correlations between NFC and tasks assumed to measure executive functioning [52]. In the present study, although latent Cognitive Functioning—being derived from the Trail-Making Test and thus targeting processing speed, working memory, and shifting ability—showed some relation to latent Demand Avoidance, it was rather low. Moreover, no latent correlation whatsoever was obtained between Cognitive Functioning and the latent personality variable, i.e., Cognitive Effort Investment.

Low interrelations among measures designed to assess executive functioning, self-control or more generally self-regulation, and between these measures and personality traits have been attributed to low reliabilities [53, 54]. The issue of low reliability mainly holds for the behavioral tasks: Hedge et al. [54] had their participants perform typical executive functioning tasks at two points in time and also assessed self-reported impulsivity measures. The mean of the intraclass correlations between the two measurement occasions reported in Tables 1 and 2 of the respective report was .56 for the executive functioning tasks and .81 for the impulsivity measure. Likewise, in a large-scale analysis of the retest reliabilities of self-regulation tasks and survey data, mean retest reliabilities of tasks vs. survey measures were .61 vs. .71 [18].

Our results mirror this picture: while the variables based on self-report exhibited a very high reliability of .98 (see Table 4), those based on cognitive tasks were lower, with .69 for the Cognitive Functioning variables and .67-.71 for the Demand Avoidance measures. Yet, when using these estimates to correct the interrelation between the measures for attenuated reliability according to the formula $r_{x'y'} = r_{xy} / \sqrt{r_{xx} * r_{yy}}$ [55]—with $r_{xy}$ being the attenuated correlation, $r_{xx}$ and $r_{yy}$ the reliabilities of the correlated variables and $r_{x'y'}$ the corrected correlation—the association remains weak, $r_{x'y'} = -.10$. This indicates, that while the issue of reliability has to be considered in correlational research, it does not explain the low effect size obtained in the present study. In our view, it is rather a conceptual issue that may account for our results.

Walter Mischel [56] was not the first to note that relationships between personality traits and actual behavior are weak at best and depend on situational variables. While under some situational conditions, individuals will more readily act in line with their stable individual patterns of behavior and experience, they will not under other conditions. "To the degree that subjects are exposed to powerful treatments, the role of individual differences will be minimized. Conversely, when treatments are weak, ambiguous, or trivial, individual differences in person variables should exert significant effects." [56]. This outlines what Mischel called *strong* and *weak* situations. Indeed, as already pointed out by Cronbach [57], in cognitive psychology, tasks are usually designed to be powerful treatments where situational variation has a strong impact on behavior, while interindividual variation is treated as noise [see also 54]. Conversely, in personality psychology, personality traits are inferred from behavioral patterns that are stable across time and situations. Here, situational variation is considered noise. Thus, in the present context, the DST may have created a rather strong situation that minimized individual differences. While interindividual variation exists, the distribution of low demand choices is shifted towards a higher propensity for demand avoidance, because the task was designed to demonstrate a general avoidance of cognitive demand. Therefore, a direct association of personality traits with behavior that draws on executive functioning may have been minimized.

### Person×situation interactions may provide one solution

In our opinion, correlational research in the context of cognitive (neuro)science therefore requires an entirely different view on what renders experimental tasks *good* tasks, i.e., tasks that systematically vary situational conditions in order to allow interindividual variation to occur. Such a perspective is explicitly taken in the person×situation interaction approach, where it is examined how situational variation and interindividual variation *interact* in the prediction of behavior [e.g., 56]. A recent theoretical model of the nature of such interactions, the *Nonlinear Interaction of Person and Situation (NIPS) Model* [58, 59] assumes that relative to a given personality trait, situational characteristics more or less afford trait-specific behavior, and that the situational affordance level interacts with the trait level in a nonlinear way (see Fig 3A). Replacing *situational affordance* by *mental demand*, *trait-specific behavior* by *mental effort expenditure* and *trait* by *trait cognitive effort investment* as measured via self-report, Fig 3B gives the prediction on the expected person×situation interaction in the present context.

To examine person×situation interactions, one would need a task where mental demand is systematically varied. Actually, the COG-ED task by Westbrook et al. [5] fulfils this requirement, because in contrast to the DST with only two demand levels, it has five to seven demand levels depending on the *n*-back level. Nevertheless, it remains to be determined whether *n*-back levels monotonically increase subjective demand or whether, at some level, individuals relinquish the task. Yet, judging from the scatter plot presented in Fig 3 in Westbrook et al. [5], the effect size for the correlation of NFC with effort discounting seems to be medium at best just as the original finding of Kool et al. [8], and to our knowledge, the replicability of this effect remains to be established [but see 60, for a children sample].

## Conclusion

The present study provides evidence that not only self-reported Cognitive Effort Investment but also behavioral Demand Avoidance are trait-like, given their substantial portions of trait variance. However, we could not establish a relationship between the trait aspects of Cognitive

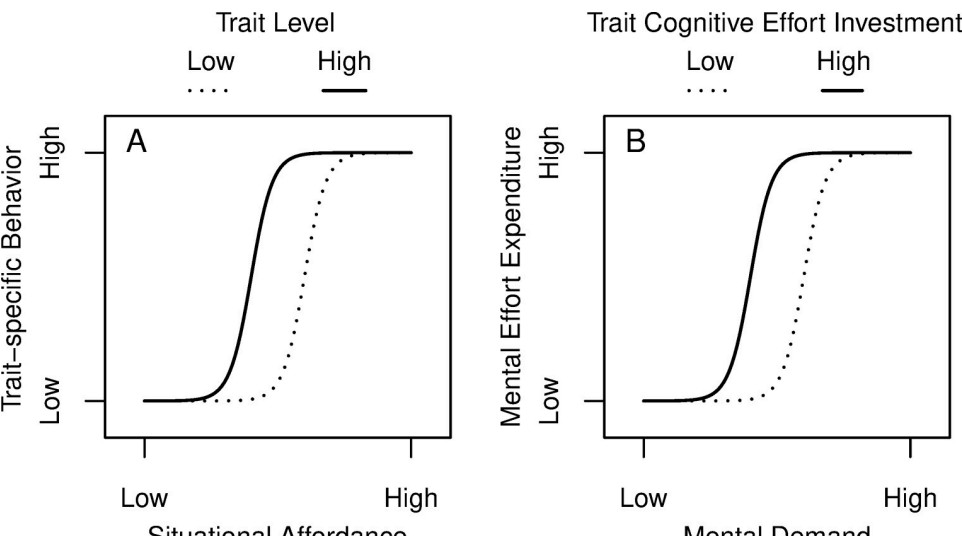

**Fig 3. Nonlinear interaction of person and situation.** (A) hypothetical interaction effect between situational affordance and trait levels on the intensity of trait specific behavior in general; (B) hypothetical interaction effect between the mental demand and trait Cognitive Effort Investment on the intensity of the expenditure of mental effort.

Effort Investment and Demand Avoidance. Moreover, the direct correlation between Self-Control and Demand Avoidance was low and insignificant as well, despite adequate power to detect an effect half of the one originally reported. Results such as ours seem to be the rule rather than the exception because overall, personality-behavior relationships can be expected to be weak at best due to the approach taken in cognitive psychology that tends to minimize interindividual variance in cognitive tasks. This renders significant personality-behavior relationships unlikely. In our view, correlational research in cognitive (neuro)science needs a fresh start, using tasks that allow for both interindividual and systematic situational variation and examining person×situation interactions. Such an approach will hopefully provide a more differentiated view on whether self-reported Cognitive Effort Investment is systematically related to actual behavioral tendencies to avoid cognitive demand.

## Supporting information

**S1 Appendix.**
(DOCX)

## Acknowledgments

We are indebted to Fanny Weber-Göricke for assistance in data acquisition management.

## Author Contributions

**Conceptualization:** Alexander Strobel, Philipp C. Paulus, Sebastian Pannasch, Stefan J. Kiebel.

**Data curation:** Gesine Wieder, Corinna Kührt.

**Formal analysis:** Alexander Strobel.

**Funding acquisition:** Alexander Strobel, Sebastian Pannasch, Stefan J. Kiebel.

**Investigation:** Gesine Wieder, Corinna Kührt.

**Methodology:** Alexander Strobel.

**Project administration:** Gesine Wieder, Corinna Kührt.

**Software:** Philipp C. Paulus.

**Supervision:** Alexander Strobel.

**Validation:** Corinna Kührt.

**Writing – original draft:** Alexander Strobel.

**Writing – review & editing:** Alexander Strobel, Gesine Wieder, Philipp C. Paulus, Florian Ott, Sebastian Pannasch, Stefan J. Kiebel, Corinna Kührt.

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
