## [Decision Letter · Decision Letter 0]

12 Feb 2020

PONE-D-19-33696

Dispositional cognitive effort investment and behavioral demand avoidance: Are they related?

PLOS ONE

Dear Dr. Strobel,

Thank you for submitting your manuscript to PLOS ONE. After careful consideration, we feel that it has merit but does not fully meet PLOS ONE’s publication criteria as it currently stands. Therefore, we invite you to submit a revised version of the manuscript that addresses the points raised during the review process.

We would appreciate receiving your revised manuscript by Mar 27 2020 11:59PM. To enhance the reproducibility of your results, we recommend that if applicable you deposit your laboratory protocols in protocols.io, where a protocol can be assigned its own identifier (DOI) such that it can be cited independently in the future. For instructions see: http://journals.plos.org/plosone/s/submission-guidelines#loc-laboratory-protocols

We look forward to receiving your revised manuscript.

Kind regards,

Valerio Capraro

Academic Editor

PLOS ONE

Additional Editor Comments (if provided):

I have now collected two reviews from two experts in the field. Both reviewers find the topic of the paper interesting, but they diverge in their final judgment, one recommends minor revision and the other one recommends rejection. After reading the reviews, I have convinced myself that this work should be given a chance. My feeling is that the comments of the negative reviewer can be potentially addressed in a major revision, primarily focused on improving the introduction and, in particular, the motivation of this work as well as the connection with prior work. Therefore, I would like to invite you to revise your paper following the reviewers' comments. Needless to say that all comments myst be addressed.

I am looking forward for the revision.

Journal Requirements:

2. Your ethics statement must appear in the Methods section of your manuscript. If your ethics statement is written in any section besides the Methods, please move it to the Methods section and delete it from any other section. Please also ensure that your ethics statement is included in your manuscript, as the ethics section of your online submission will not be published alongside your manuscript.

3. We noted in your submission details that a portion of your manuscript may have been presented or published elsewhere. ["The sample used for the study described in the manuscript is a subsample of the sample used in in a study that preceded the present one"] Please clarify whether this publication was peer-reviewed and formally published. If this work was previously peer-reviewed and published, in the cover letter please provide the reason that this work does not constitute dual publication and should be included in the current manuscript.

Reviewers' comments:

Reviewer's Responses to Questions

**Comments to the Author**

1. Is the manuscript technically sound, and do the data support the conclusions?

Reviewer #1: Partly

Reviewer #2: Yes

2. Has the statistical analysis been performed appropriately and rigorously? 

Reviewer #1: I Don't Know

Reviewer #2: Yes

3. Have the authors made all data underlying the findings in their manuscript fully available?

Reviewer #1: Yes

Reviewer #2: Yes

4. Is the manuscript presented in an intelligible fashion and written in standard English?

Reviewer #1: Yes

Reviewer #2: Yes

5. Review Comments to the Author

Reviewer #1: In this submission, Strobel and colleagues examine, using a factor analysis, the possible relationships between self-reported motivation to exert effort (i.e. NFC), self-control, and ‘intellect,’ (collected at multiple timepoints) and task-based measures: the DST and two versions of an established trail-making test, all given at They find, applying a CFA, that a stable trait-based factor explains self-reported demand avoidance (even controlling for differences in cognitive ability), but overall weak relationships between self-report measures and actual task behavior.

Admittedly, I am not a personality researcher and the statistical methods employed here (e.g., CFA) would be better evaluated by someone from this research community. However, as someone with a strong interest in cognitive effort, I found myself wondering why this study was done-- what particular gap in the effort literature does this work address? Perhaps this is a matter of the authors providing a better motivation for their efforts, but the best I could find was: “to determine the trait-like nature of both self-reported Effort investment and behavioral Demand avoidance in order to systematically relate stable individual differences in these measures to each other” (p 7). Why is this important to do? For example, for cognitive effort researchers, what exactly is staked on the validity of these self-report measures? Would any researcher whole relies primarily upon behavioral or task-based measures—rather than self-report—be surprised at the lack of relationship ? Overall, I found the introduction to be lengthy but I found that they key motivation for this study (and accordingly, its possible contribution within the literature) gets a bit ‘lost in the shuffle.’

Other points:

- Oddly one of the key task-based measures noted in the introduction (p 1) is the COG-ED. Yet The authors report that they refrained from using the COG-ED measure (a somewhat well-established measure of effort avoidance vis-à-vis discounting) because of “the rather low correspondence to be expected of this measure and the DST and instead employed two versions of the DST” (p 6). This analysis doesn’t appear to be reported anywhere in this submission (I assume the authors did it in some prior version of this study?), but getting COG-ED choice data seems important given the importance placed on this measure at numerous points in the manuscript. Isn’t it documented by Westbrook et al. (2013) that NFC correlates well with COG-ED-assessed discounting rates

- The authors contend this work is a conceptual replication of Kool et al. 2013. In what way is this a conceptual replication of this study? Their work presents an analysis of neuroimaging/behavioral data. The connection between this study and their is not made readily apparent to readers.

Reviewer #2: This is a well-conducted study and the paper should be published. It finds that both self-reported Cognitive Effort Investment and behavioral Demand Avoidance are stable, trait-like measures, but they are not correlated. The latter result is surprising, given the conceptual relatedness and also prior research that sometimes finds this correlation. However, the methodology used to derive this result is sound and its interpretation in terms of person-situation interactions is convincing.

I have a couple of questions that the authors need to be answer before I can recommend the paper for publication.

1. I find it surprising that the new measure of demand avoidance (i.e., the one that considers demand detection) does not seem to differ much from the old measure of demand avoidance. What was the procedure for determining the demand detection point?

2. In the section “Personality-behavior relationships are weak at best”, last paragraph, the authors mention “the task was designed to demonstrate—and actually instructed to induce—the avoidance of cognitive demand”. Please clarify what you meant by “actually instructed to induce”, because as far as I know most studies using DST (including Kool’s original ones) do not attempt to bias subjects toward demand avoidance.

6. PLOS authors have the option to publish the peer review history of their article (what does this mean?). If published, this will include your full peer review and any attached files.

Reviewer #1: No

Reviewer #2: Yes: Ion Juvina

---

## [Author Response · Author response to Decision Letter 0]

26 Feb 2020

In the following, we provide authors’ responses (AR) for all points raised by the editorial office (EP) and two reviewers (RP). We consider the questions raised by the former serious and believe that we could convincingly answer them. We also greatly appreciate the comments of the latter and hope that we were able to adequately address all the issues raised. Changes to the manuscript/supplemental material are highlighted by red font.

Editorial office

EP: Did the authors present any new data in this submission that were not previously presented in the published article Kührt et al. (2019)? Did the authors perform any additional experiments or collect any additional data that were not a part of the study from the published article Kührt et al. (2019)?

AR: A clear yes to both your questions. We present new data, did perform additional experiments and collected additional data that were not part of the manuscript by Kührt et al. (2019), which is presently only openly available as a preprint on PsyArXiv (https://psyarxiv.com/rwz62/), but is under review at another journal. We apologize for obviously not having acknowledged clearly enough the relation of our submission to the manuscript by Kührt et al. (2019). In our submission, we wrote (p. 8, lines 172-176): “Please note that this sample is a subsample of the sample used in Study 2 in Kührt et al. [22] where we replicated the hierarchical factor model of Cognitive Effort Investment. The present sample consists of those participants who took part in both assessments and had complete data for all measures relevant for the present report. There is no duplicate reporting of results.” 

In more detail: Both manuscripts are part of a larger project on individual differences in effort investment and adjustments in cognitive control. Study 1 in Kührt et al. (2019) was the first study of this project, our manuscript is about the second study of this project. In Kührt et al. (2019), we aimed at establishing a factor model of Cognitive Effort Investment that integrated the traits Need for Cognition and Self-Control in one overarching construct. This laid groundwork for the second study described in the present submission, where we aimed at relating behavioral demand avoidance to one overarching construct, i.e., Cognitive Effort Investment, instead of two or even more, thereby potentially increasing type I error rates. Thus, although there is a common theme across both manuscript, they differ with regard to data sets, research questions and analyses. Merging both manuscripts would have resulted in a very long and potentially inaccessible paper about two different research questions.

With regard to the sample used in our study, we had a sample of participants who came to our lab twice to complete personality questionnaires and performed two demand selection tasks as well as a cognitive task battery at both time points, with an interval between the assessments of about five weeks. At the first assessment, N = 244 participants took part, but only N = 217 took part in the second assessment. In Kührt et al. (2019), we used all participants who had taken part in the first assessment and only the personality data from this first assessment to replicate the integrative factor model of Cognitive Effort Investment established in a preceding study using Confirmatory Factor Analysis (CFA) of data from an unrelated larger sample assessed online. In the present study, we used a subsample of participants who had taken part in both assessments and used the personality data to relate them to the behavioral data from the experimental tasks, controlling for cognitive functioning measures derive from the cognitive task battery. The method used for this was latent state-trait modeling. Thus, both with regard to the data used and the methods applied, we present new data. Only for the purpose of deriving personality factor scores for latent state-trait modeling, we referred to the CFA method, using less participants, but more data, i.e., from both time points, in one CFA model. Accordingly, in the present submission, both the CFA models fitted and the results reported for factor loadings (S1 Appendix: Figure S2) and model fit (p. 19, lines 408-409) differ from that reported in Kührt et al. (2019), and are also not of primary interest, but provided for reasons of completeness only. 

We hope we could make clear that while having one common theme—dispositional cognitive effort investment—both studies target different research questions, primarily use different methods and that our submission presents new data, analyses and results not published previously. We have included the content of the previous paragraph in the Supplementary Material under Supplementary Methods (and refer to this in the main manuscript, p. 8, line 176) to be as transparent as possible also to all readers interested in our study.

Reviewer #1

RP1.1: Admittedly, I am not a personality researcher and the statistical methods employed here (e.g., CFA) would be better evaluated by someone from this research community. However, as someone with a strong interest in cognitive effort, I found myself wondering why this study was done— what particular gap in the effort literature does this work address? Perhaps this is a matter of the authors providing a better motivation for their efforts, but the best I could find was: “to determine the trait-like nature of both self-reported Effort investment and behavioral Demand avoidance in order to systematically relate stable individual differences in these measures to each other” (p 7). Why is this important to do? For example, for cognitive effort researchers, what exactly is staked on the validity of these self-report measures? Would any researcher whole relies primarily upon behavioral or task-based measures—rather than self-report—be surprised at the lack of relationship ? Overall, I found the introduction to be lengthy but I found that they key motivation for this study (and accordingly, its possible contribution within the literature) gets a bit ‘lost in the shuffle.’

AR1.1: We agree with the reviewer that from the original introduction it can be difficult to understand the purpose of the study. We apologize for not having motivated clearly the point of our study. We aimed at validating an often-made assertion in the literature. This assertion is that self-reported self-control (and similar traits related to dispositional effort investment) are related to behavioral demand avoidance. In our view, this is important because it is all too often assumed that measuring a personality characteristic at one point in time informs about the trait level of that characteristic and, critically, that correlates of that characteristic will also be trait-like. For example, in their highly influential paper Westbrook et al. (2013) state that “lesser effort discounting predicted greater self-reported engagement with cognitively demanding activities using the NCS. This correlation supports: 1) a trait-like property of effort costliness […] and 2) that our measure captures that trait.” (Westbrook et al., 2013, p. 7). In the field of interindividual differences research, this is quite a critical statement with potentially far-reaching consequences: If this were the case, then the behavioral measure may qualify as a time-stable predictor for real-life outcomes. However, it could also be the case that the behavioral measure just happened to be correlated with the personality measure for some state influences at the point in time both were measured, which would compromise the predictive nature of the behavioral measure.

Although Westbrook et al. (2013) and Kool et al. (2013) explicitly and implicitly argue that the correlations observed (between Need for Cognition and effort discounting as in Westbrook et al., 2013, or between Self-Control and demand avoidance as in Kool et al., 2013) point to some trait-like behavioral characteristic in the measures obtained via the COG-ED paradigm or the demand selection task (DST), this trait-likeness has not been explicitly shown yet. To establish whether performance in these tasks is trait-like, we determined the trait-like nature of at least one measure of demand avoidance/effort discounting (see AR1.2) and related the stable trait-variance in these measures to the stable trait variance in self-reported cognitive effort investment. Our aims were thus two-fold: First, to establish that behavioral demand avoidance was trait-like at all, and second, to test whether trait variance covaries with the trait variance in self-reported cognitive effort investment. 

We believe our findings to be valuable for a wider community of cognitive effort researchers. To derive predictions about real-life outcomes from some experimental measures, it would be highly relevant to relate the experimental measures to self-report data of typical effort-related behavior in situations that require the exertion of mental effort. To do so validly, we believe that it is also of interest to have some proxies for the stable, trait-like nature of both the behavioral measures and the self-report data that may be related to the former. Our results show that measures of behavioral demand avoidance display enough stable trait variance to be of importance as predictors of real-life outcomes, but that – at least in our experiments – there is no evidence for trait-covariation between behavioral and self-report measures. 

Of course, one may question whether self-report and behavioral data need to show some correspondence. We firmly believe that they should, because otherwise one could not draw reliable conclusions on behavioral patterns originating from assumed stable individual differences. In summary, we agree with the reviewer that we did not made clear enough the intention and therefore the novelty of our study and now properly motivate our study in the introduction (pp. 6-7, lines 114-143). 

RP1.2: Oddly one of the key task-based measures noted in the introduction (p 1) is the COG-ED. Yet The authors report that they refrained from using the COG-ED measure (a somewhat well-established measure of effort avoidance vis-à-vis discounting) because of “the rather low correspondence to be expected of this measure and the DST and instead employed two versions of the DST” (p 6). This analysis doesn’t appear to be reported anywhere in this submission (I assume the authors did it in some prior version of this study?), but getting COG-ED choice data seems important given the importance placed on this measure at numerous points in the manuscript. Isn’t it documented by Westbrook et al. (2013) that NFC correlates well with COG-ED-assessed discounting rates.

AR1.2: We thank the reviewer for bringing up this key point. Indeed, we originally intended to use both the Demand Selection Task (DST) and the COG-ED paradigm as two measures of the same construct. In preparation and construction of the study we faced several points in favor of implementing only the DST:

(1) To have two measures of the same construct was important for our aims, as latent state-trait (LST) modeling requires to have (at least) two measures of one construct at each time point for estimating the state-, method- and trait-specific variances in measures of the construct. This implies that the measures of the construct in question should not be too different. Given that the DST and the COG-ED have a quite different structure (demand avoidance vs. effort discounting), we reasoned that we would risk to have too low an overlap of both tasks (i.e., behavior in these tasks may be not as highly correlated as required for LST modeling). 

(2) We actually piloted both tasks in a number of participants, however, in a between design, which is why we do not have own data on a direct comparison between the DST and the COG-ED. Yet, valid conclusions about the existence of a correlation of demand avoidance in the DST with effort discounting in the COG-ED of, say, r ≥ .30 would have required a within-design with at least 80 participants, a piloting effort that was simply not manageable. 

(3) We therefore had to decide whether we could construct two similar versions of the DST or the COG-ED and found the DST task more promising and manageable than the COG-ED task. 

(4) In addition, during piloting, participants had expressed great discomfort with the COG-ED. Since we were dependent on the participants coming to the laboratory twice and did not drop out because of disliking the task, it seemed more sensible to us to use two variants of the DST. 

Interestingly, our decision was reinforced when we learned during informal talks at conferences that apparently, the DTS and the COG-ED are not substantially correlated to each other. Indeed, in hindsight, we might have arrived at other conclusions had we used two variants of the COG-ED. Still, as stated in the Discussion section of the manuscript, one has to consider that the correlation between effort discounting and Need for Cognition observed by Westbrook et al. (2013) remains to be replicated, and so far, we found only one replication attempt in a children sample. In our opinion, our theoretical and methodological consideration of the Westbrook et al. (2013) study and their COG-ED paradigm in the current version of the main manuscript addresses the points outlined above, but still holds a balance between informing the readers and avoiding too detailed information on a task not used. For those interested in this issue in more detail, we elaborate on these reasons, as outlined above, in the Supplemental Material (S1 Appendix: Supplementary Methods, pp. 2-3). 

RP1.3: The authors contend this work is a conceptual replication of Kool et al. 2013. In what way is this a conceptual replication of this study? Their work presents an analysis of neuroimaging/behavioral data. The connection between this study and their is not made readily apparent to readers.

AR1.3: We apologize that we did not make this clear enough. A prominent part of the work of Kool et al. (2013) is about a synthesis of neuroimaging data on self-control and cognitive costs. In the empirical part, however, Kool et al. (2013) show demand avoidance to be related to self-reported self-control. It is this specific result that we aimed at replicating conceptually. Apart from also providing a direct replication (i.e., the bivariate correlation of self-control and the demand avoidance measure used by Kool et al., 2013, see p. 21 of our manuscript), we conceptually replicate the result by using a more general measure of dispositional cognitive effort investment and a more general measure of demand avoidance derived from two similar tasks. In order to make our approach clearer, we rephrased and clarified our research aim (p. 7, lines 140-143).

Reviewer #2

RP2.1: I find it surprising that the new measure of demand avoidance (i.e., the one that considers demand detection) does not seem to differ much from the old measure of demand avoidance. What was the procedure for determining the demand detection point?

AR2.1: We thank the reviewer for making this point as it allows us to clarify the procedure used. As detailed in the Supplemental Material (see S1 Appendix: Supplementary Methods, pp. 5-6), we closely followed the reviewer’s procedure: Following the suggestions made by Juvina et al. (2018), we first determined the block-wise demand detection point (DDP) by employing a sliding window of size � = 12 trials across individual decisions for patterns associated with high vs. low demand trial by trial starting with trial twelve. Using Wilcoxon sign tests, it was determined whether the individual choice rate in the respective window was significantly different from 0.5. If the Wilcoxon test was significant for a given trial and remained significant for all further trials, the DDP was set to the respective trial number. If the test yielded non-significant results for any further trial, the search for a DDP after that trial was started, and if a DDP could not be found throughout a given block, it was set to the highest trial number, i.e., 72. Choice after detection (CAD) was then determined as the percentage of low demand choices in the window from the DDP to the end of each block. In cases where the DDP was equal to the highest trial number, CAD was set to the total percentage of low demand choices across each block. The new demand avoidance measure was then determined as DDP – CAD, with DDP being the trial number where demand detection occurred and CAD being the rate of high-demand choices in the trials that followed the DDP. Both DDP and CAD were normalized by dividing the DDP by the total trial number per block, and by subtracting the rate of high-demand choices from one, respectively. The resulting new demand avoidance measure therefore ranged from -1 to 1. 

This new demand avoidance measure was correlated at rs ≥ .50 with the original measure (see S1 Appendix: Supplementary Results, p.9 Figure S3). In order to make sure that the resulting measure was not different when using a different sliding window, we also performed new analyses with other sizes of the sliding window, n = 9 and n = 18. This did not essentially impact on the results (i.e., the correlations between changed only at the second or third digit after the decimal and did not reach significance in any case). This information is now given as footnote to the procedure for determining the demand detection point (see S1 Appendix: Supplementary Methods, p. 5).

RP2.2: In the section “Personality-behavior relationships are weak at best”, last paragraph, the authors mention “the task was designed to demonstrate—and actually instructed to induce—the avoidance of cognitive demand”. Please clarify what you meant by “actually instructed to induce”, because as far as I know most studies using DST (including Kool’s original ones) do not attempt to bias subjects toward demand avoidance.

AR2.2: We thank the reviewer for giving us the opportunity to elaborate on this issue. Indeed, in Experiment 1 of the paper by Kool et al. (2010), the instruction was as follows: “Subjects were told that they were free to choose from either deck on any trial and that they should ‘feel free to move from one deck to the other whenever you choose’ but also that ‘if one deck begins to seem preferable, feel free to choose that deck more often’.” (Kool et al., 2010, p. 667). In Kool et al. (2013), a similar instruction was given: “Participants were told they were free to sample from either target, but that if they developed a preference they should feel free select one more than the other.” (Kool et al., 2013, p. 4). During initial pilot experiments, we actually used a more neutral instruction and could not find a clear demand avoidance pattern. This is consistent with evidence from your own research (Juvina et al., 2018) where demand avoidance was less pronounced when using an instruction that did not direct participants’ attention towards a difference between the choice options or indirectly suggested to choose one option more often. In contrast, when adhering to the instruction of Kool et al. (2010) in further pilot experiments, we were able to find a bias towards low demand choices and therefore used it in the main experiment. We elaborate on this issue in the revised version of the manuscript (Methods section, p. 11, lines 231-235, and Discussion section, p. 30, lines 632-633).

---

## [Decision Letter · Decision Letter 1]

13 May 2020

PONE-D-19-33696R1

Dispositional cognitive effort investment and behavioral demand avoidance: Are they related?

PLOS ONE

Dear Dr. Strobel,

Thank you for submitting your manuscript to PLOS ONE. After careful consideration, we feel that it has merit but does not fully meet PLOS ONE’s publication criteria as it currently stands. Therefore, we invite you to submit a revised version of the manuscript that addresses the points raised during the review process.

We would appreciate receiving your revised manuscript by Jun 27 2020 11:59PM. To enhance the reproducibility of your results, we recommend that if applicable you deposit your laboratory protocols in protocols.io, where a protocol can be assigned its own identifier (DOI) such that it can be cited independently in the future. For instructions see: http://journals.plos.org/plosone/s/submission-guidelines#loc-laboratory-protocols

We look forward to receiving your revised manuscript.

Kind regards,

Valerio Capraro

Academic Editor

PLOS ONE

Additional Editor Comments (if provided):

The reviewers are happy with the revision, but one of them suggests some minor changes before publication. Please address these remaining issues at your earliest convenience. I am looking forward for the final version.

Reviewers' comments:

Reviewer's Responses to Questions

**Comments to the Author**

1. If the authors have adequately addressed your comments raised in a previous round of review and you feel that this manuscript is now acceptable for publication, you may indicate that here to bypass the “Comments to the Author” section, enter your conflict of interest statement in the “Confidential to Editor” section, and submit your "Accept" recommendation.

Reviewer #1: (No Response)

Reviewer #2: All comments have been addressed

2. Is the manuscript technically sound, and do the data support the conclusions?

Reviewer #1: Yes

Reviewer #2: (No Response)

3. Has the statistical analysis been performed appropriately and rigorously? 

Reviewer #1: (No Response)

Reviewer #2: (No Response)

4. Have the authors made all data underlying the findings in their manuscript fully available?

Reviewer #1: Yes

Reviewer #2: (No Response)

5. Is the manuscript presented in an intelligible fashion and written in standard English?

Reviewer #1: Yes

Reviewer #2: (No Response)

6. Review Comments to the Author

Reviewer #1: I still believe more work needs to be done in the manuscript to address my previous comment (AR1.1)—the authors’ response does a good job clarifying the purpose of this study but the revised Introduction doesn’t really make this underlying motivation clear. To this end, I think the introduction would benefit from some restructuring. I was also confused by the mention of ‘real-life outcomes’ in the added text to page 7. What are these real-life outcomes and how can we measure them?—trait measures, task-based measures, or neither?

Reviewer #2: (No Response)

7. PLOS authors have the option to publish the peer review history of their article (what does this mean?). If published, this will include your full peer review and any attached files.

Reviewer #1: No

Reviewer #2: Yes: Ion Juvina

---

## [Author Response · Author response to Decision Letter 1]

11 Sep 2020

Reviewer #1

RP1.1: I still believe more work needs to be done in the manuscript to address my previous comment (AR1.1)—the authors’ response does a good job clarifying the purpose of this study but the revised Introduction doesn’t really make this underlying motivation clear. To this end, I think the introduction would benefit from some restructuring. I was also confused by the men-tion of ‘real-life outcomes’ in the added text to page 7. What are these real-life outcomes and how can we measure them?—trait measures, task-based measures, or neither?

AR1.1: We thank the reviewer for the opportunity to further improve the structure of the Intro-duction and clarify thereby the underlying motivation. We agree that in the previous version of the Introduction, the actual motivation for the study becomes only apparent on page 6, which seems too late. 

In the revised version, we state our research question already in the first paragraph of p. 4. We then continue with discussing the problem that the relation of two putative traits cannot be de-termined using only measures per trait at one point in time, but rather using latent state-trait modeling (the remainder of p. 4). However, to do so, one needs two parallel measures each per construct – here self-reported cognitive effort investment and behavioral cognitive effort avoid-ance (pp. 4-5). We then outline the parallel behavioral measures originally intended and after-wards consider the question of whether NFC and Self-Control are parallel self-report measures of cognitive effort investment (pp. 5-7).

We believe that in this revised form, the purpose of our study becomes more obvious early in the Introduction. We hope that readers with no background in personality and individual differ-ences will find this structure an easier access to our research. 

Regarding ‘real-life outcomes’ and how to measure them, we detailed this by rewriting the re-spective sentence (now p. 4, lines 66-69): “If so, then such behavioral measures would qualify as time-stable predictors for real-life outcomes such as self-control failures in everyday life as measured via ecological momentary assessment (e.g., [12]) or relapse from addiction treatment (e.g., [13]). The cited references give examples of studies that used behavioral measures as pre-dictors of real-life outcomes: Wolff et al. (2016) found that behavioral measure of executive function predict real-life self-control failures, and Kräplin et al. (2020) observed that behavioral measures impulsive decision-making predicts the course of addictive disorders. 

Reviewer #3

RP3.1: The results reported here are, admittedly, a bit disappointing for me, but they are not surprising. I spoke with the lead author at a conference a while ago, and we talked about this work. I pointed him towards the recent work by Juvina, who computed an alternative measure of demand avoidance in the DST, but unfortunately this seems to not have lead to any qualita-tive changes in the results. Regardless of my personal feelings, I think the authors have per-formed a valiant effort and I applaud them for their experimental and statistical rigor.

AR3.1: We thank again for this comment and thank the reviewer very much for his overall as-sessment of our work. 

RP3.2: The only difference I can see between our implementation of the DST and the version that this team uses (aside from the consonant/vowel - capital/non-capital manipulation), is that we deliberately never tell our participants how well they are performing. As we detailed in the 2010 paper that introduced this task, we never do this because we want to avoid that participants are using an error-avoidance rather than an effort-avoidance strategy. In fact, the 2010 paper describes some pain-staking efforts to rule out that participants’ behavior on this task is driven by a desire to reduce error commission. Therefore, I was surprised to see that this team did not only include accuracy feedback at the end of each block, but also that they provided encourage-ment either affirming that their current behavior was desirable or pushing for a change. It is therefore possible that in the current version of the DST the choice behavior can’t be seen as a pure measure of pure effort avoidance, but that it reflects error avoidance (or a desire to perform well) instead. It is important to note that effort and error avoidance are not necessarily related to each other. Therefore, I think that, at the very least, the section named “self-reported cognitive effort investment and behavioral demand avoidance are unrelated” should include a discussion of this possibility.

AR3.2: We thank the reviewer for this very insightful comment. Yes, it seems we did not con-sider a possible bias arising from performance feedback/encouragement. To address the possi-bility that choice behavior in our implementation of the DST was to some extent driven by an error avoidance strategy, we reanalyzed our data as follows: For each participant, DST version, and time point, we predicted the average choice behavior across blocks 2 to 8 by the average hit rate during the preceding block by means of a linear mixed model, allowing for random inter-cepts and slopes per individual. Hit rates during the previous block did not significantly predict choice behavior in the current block (all p > .182). We now report these results in the Results section (lines 443-449) and, as the reviewer suggests, we also discuss this issue in the Discus-sion section under “Self-reported cognitive effort investment and behavioral demand avoidance are unrelated” (lines 585-591). Even in the absence of a significant effect of previous perfor-mance on current choice behavior, it remains a limitation that we did not establish a task envi-ronment identical to that of the original DST.

RP3.3: Related to this point I want to mention that when we run the task, we always aim to not push people in any direction, only telling them that they can sample freely between the decks, but that they can choose one more if the prefer it. Note that this latter instruction doesn’t neces-sarily push them towards the least effort option, as the authors seem to insinuate. It is complete-ly possible that participants take it as meaning they can choose cues based on their location or visual appearance. In contrast, the current version of this task does provide some signal that people could interpret as a push to change their behavior in a consistent way (through the accu-racy feedback).

AR3.3: We thank the reviewer for pointing out this difference to the original task. Yet, we do think that the original instruction “if one deck begins to seem preferable, feel free to choose that deck more often” (Kool et al., 2010, p. 667) may be taken by some participants as some sugges-tion to behave in a certain way. This notion is fostered by our own pilot experiments and by the results of study 1 by Juvina et al. (2018): with a more neutral instruction, demand avoidance was less pronounced. Still, we agree with the reviewer that the original instruction does not nec-essarily push participants to choose the low effort option more often as one sentence in our Dis-cussion section suggested. It originally read: “While interindividual variation exists, the distribu-tion of low demand choices is shifted towards a higher propensity for demand avoidance, be-cause the task was designed to demonstrate a general avoidance of cognitive demand and the original instruction even suggests that one should feel free to choose a preferable deck more often.” While the reviewer surely agrees to the first part of the sentence, the second (underlined) part of the sentence is a debatable interpretation. We therefore deleted the underlined part in our revision (p. 29, lines 639-641).

RP3.4: Finally, I found this paper rather hard to read. Upon some reflection, this is primarily caused by the excessive use of abbreviations, acronyms and initialisms (especially in the tables and figures). I simply do not have the working memory capacity needed to maintain them all (and I think most other readers do not have either). Would the paper be hurt by spelling out var-iables by their full name? To be honest, I think this would greatly improve readability. For ex-ample, the terms “reliability, trait consistency, etc.” are not named very often, so I am not sure they need to have an abbreviation. Moreover, if the authors end up deciding to include some of them, there should be more consistency. Sometimes variables for demand avoidance start with only a D (e.g., DMP2), and at other times with DA (e.g., DAo). Anything that can be done here would be greatly appreciated.

AR3.4: We apologize for our inflationary use of abbreviations and tried our best to remove any unnecessary abbreviations from the manuscript. In all tables, the variables are now spelled out, the abbreviations for reliability, trait consistency etc. are removed as well. Only in the figures, we kept abbreviations due to limitations in space.

RP3.5: Line 187 - The variable intellect seems to not measure intellect per se (“the faculty of reasoning and understanding objectively, especially with regard to abstract or academic mat-ters”), but rather something akin to need for cognition. I think the term used for this measure should reflect this (e.g., “interest in intellect”, “motivation for intellect”).

AR3.5: We fully agree with the reviewer that the term “intellect” may not be the best label for the trait measure in question. Yet, this term was not coined by us, but was used by Mussel (2013) in his effort to provide a framework that structures traits related to intellectual engage-ment and curiosity. The term “intellect” traces back to a distinction of two aspects of one of the traits of the five factor model of personality (e.g., McCrae & Costa, 1997), Openness to Experi-ence. DeYoung et al. (2007) proposed that each of the five factors of personality is composed of two aspects. For Openness to Experience, they proposed that these aspects are Openness and Intellect. Openness in the narrow sense refers to the aspect of the broader concept of Openness to Experience that captures individual differences related to open-mindedness with regard to aesthetics, imagination and fantasy. Intellect on the other hand refers to ingenuity and openness to ideas or values, and this term has been used in some of the progenitors of the currently ac-cepted five factor model of personality. We are afraid that elaborating on this issue would go beyond the scope of our manuscript, and we do not see us in the position to rename this variable for our purposes. 

RP3.6: Is the variable pertaining to cognitive functioning made up completely of the Trail Mak-ing Test? Wouldn’t be better to name this measure something closer to that variable? Given its broad name, it now seems to measure something more general.

AR3.6: We agree that the label “cognitive functioning” leaves the impression of a far more gen-eral measure of cognitive ability that we actually have when using only two versions of the Trail Making Test. Indeed, we originally aimed at having a broader measure of cognitive ability de-rived not only from the Trail Making Test, but also from other measures such as the Digit Span Backwards and the Digit-Symbol Substitution tests taken from the Wechsler Adult Intelligence Scale III. Yet, it turned out that the indices of cognitive ability derived from the latter tests did not correlate as strongly with each other and with the indices of the two Trail Making Test ver-sions as expected. It was therefore not possible to use all measures to create two comparable measures of cognitive functioning that we needed for latent state-trait modeling (see the Sup-plementary Material for details). Actually, we had some debate on how to term this variable given that it eventually was only composed of the two Trail Making Test versions. Yet, the Trail Making Test version A measures mental speed and to some extent working memory, while the Trail Making Test version B in addition requires shifting, and hence, central cognitive correlates of general mental ability are captured by this task. Therefore, we finally decided to use the term “cognitive functioning” to refer to a variable that clearly captures central aspects of general men-tal ability, but does not evoke the idea of a broader concept like the original intended “cognitive ability”. We hope that it now becomes clearer why we chose this very term. 

RP3.7: Line 336 - What is “demand avoidance cognitive functioning”?

AR3.7: Thank you for noticing this. This actually was a mistake, it should have read “demand avoidance and cognitive functioning”. We corrected it in the revised version of this manuscript. 

References

DeYoung, C. G., Quilty, L. C., & Peterson, J. B. (2007). Between facets and domains: 10 aspects of the Big Five. Journal of Personality and Social Psychology, 93(5), 880-896.

Kool, W., McGuire, J. T., Rosen, Z. B., & Botvinick, M. M. (2010). Decision making and the avoidance of cognitive demand. Journal of Experimental Psychology: General, 139(4), 665-682.

Juvina, I., Nador, J., Larue, O., Green, R., Harel, A., & Minnery, B. (2018). Measuring individual differences in cognitive effort avoidance. In T. Rogers, M. Rau, X. Zhu, & C. W. Kalish (Eds.), Proceedings of the 40th Annual Conference of the Cognitive Science Society (pp. 1886-1891). Austin, TX: Cognitive Science Society.

McCrae, R. R., & Costa, P. T. (1997). Personality trait structure as a human universal. Ameri-can Psychologist, 52(5), 509-516.

Mussel, P. (2013). Intellect: a theoretical framework for personality traits related to intellectual achievements. Journal of Personality and Social Psychology, 104(5), 885-906.

---

## [Editor Report · Decision Letter 2]

15 Sep 2020

Dispositional cognitive effort investment and behavioral demand avoidance: Are they related?

PONE-D-19-33696R2

Dear Dr. Strobel,

We’re pleased to inform you that your manuscript has been judged scientifically suitable for publication and will be formally accepted for publication once it meets all outstanding technical requirements.

Kind regards,

Valerio Capraro

Academic Editor

PLOS ONE
---

## [Editor Report · Acceptance letter]

28 Sep 2020

PONE-D-19-33696R2 

Dispositional cognitive effort investment and behavioral demand avoidance: Are they related? 

Dear Dr. Strobel:

I'm pleased to inform you that your manuscript has been deemed suitable for publication in PLOS ONE. Congratulations! Your manuscript is now with our production department. 

Kind regards, 

on behalf of

Dr. Valerio Capraro 

Academic Editor

PLOS ONE